# Reverse dark current in organic photodetectors and the major role of traps as source of noise

Jonas Kublitski [1✉], Andreas Hofacker [1✉], Bahman K. Boroujeni [2,3], Johannes Benduhn [1], Vasileios C. Nikolis [1,4], Christina Kaiser[5], Donato Spoltore[1], Hans Kleemann[1], Axel Fischer[1], Frank Ellinger[2,3], Koen Vandewal [6✉] & Karl Leo[1,3]

Organic photodetectors have promising applications in low-cost imaging, health monitoring and near-infrared sensing. Recent research on organic photodetectors based on donor–acceptor systems has resulted in narrow-band, flexible and biocompatible devices, of which the best reach external photovoltaic quantum efficiencies approaching 100%. However, the high noise spectral density of these devices limits their specific detectivity to around $10^{13}$ Jones in the visible and several orders of magnitude lower in the near-infrared, severely reducing performance. Here, we show that the shot noise, proportional to the dark current, dominates the noise spectral density, demanding a comprehensive understanding of the dark current. We demonstrate that, in addition to the intrinsic saturation current generated via charge-transfer states, dark current contains a major contribution from trap-assisted generated charges and decreases systematically with decreasing concentration of traps. By modeling the dark current of several donor–acceptor systems, we reveal the interplay between traps and charge-transfer states as source of dark current and show that traps dominate the generation processes, thus being the main limiting factor of organic photodetectors detectivity.

[1] Dresden Integrated Center for Applied Physics and Photonic Materials (IAPP) and Institute for Applied Physics, Technische Universität Dresden, Nöthnitzer Str. 61, 01187 Dresden, Germany. [2] Chair of Circuit Design and Network Theory (CCN), Technische Universität Dresden, 01069 Dresden, Germany. [3] Center for Advancing Electronics Dresden (cfaed), Technische Universität Dresden, 01062 Dresden, Germany. [4] Heliatek GmbH, Treidlerstrasse 3, 01139 Dresden, Germany. [5] Swansea University, Singleton Park SA2 8PP Wales, UK. [6] Instituut voor Materiaalonderzoek (IMO), Hasselt University, Wetenschapspark 1, BE-3590 Diepenbeek, Belgium. ✉email: jonas.kublitski@tu-dresden.de; andreas.hofacker@tu-dresden.de; koen.vandewal@uhasselt.be

Light sensing and imaging[1] are important technological fields and create high demand for photodetectors (PDs). Besides a high responsivity, a low-noise spectral density ($S_n$), resulting in a high specific detectivity ($D^*$), is a key requirement. Currently, PDs for the visible and near-infrared spectral region are mainly based on silicon (Si) and indium gallium arsenide (InGaAs) alloys. While their performance is outstanding, devices and imagers are expensive and inflexible. On the other hand, organic photodetectors (OPDs) can be significantly cheaper, but these devices still suffer from a high $S_n$, resulting in rather low detectivities. Among the many sources of noise, the shot noise, proportional to the dark current, has been suggested to play a major role[2], especially because OPDs usually operate in reverse bias voltages, where the measured reverse dark current ($J_D$) strongly deviates from its ideal value.

Dark current suppression in organic diodes has been the subject of several reports in the literature[3]. Most frequently used approaches are charge selective layers[4,5], contact alignment[6,7], prevention of shunt paths via layer thickness increase[8], and interlayers to smoothen the bottom contact[9,10], as well as charge transport layer structuring[11]. While the above-mentioned $J_D$ suppression approaches lead to an improved OPD performance, a comprehensive understanding of the intrinsic and extrinsic sources of dark current is still missing, which would provide insights for future device optimization using improved materials or architectures.

In an ideal diode, in addition to the diffusion current, the dark saturation current ($J_0$) comprises a thermally activated component as a result of thermal generation of charges over the gap of the material[12]. However, in organic diodes formed by a donor–acceptor (D–A) structure, charge-transfer (CT) states are present at the interface[13]. Being usually lower than the gap of the single components, the effective gap of the blend is the characteristic charge-transfer state energy ($E_{CT}$). Therefore, the activation energy of the ideal dark current of organic diodes, based on D–A blends, is determined by $E_{CT}$.

In this work, we show that $J_D$ indeed scales with $E_{CT}$ and values as low as $10^{-7}$ mA cm$^{-2}$ are achieved for an $E_{CT}$ of 1.58 eV at −1 V. However, the measured $J_D$ is orders of magnitude higher than the ideal, thermally generated dark current, $J_0$, calculated within the radiative limit[14]. This discrepancy is commonly observed in OPDs and is the main limiting factor for achieving higher detectivities. By employing drift-diffusion simulations, we show that these higher $J_D$ values can be explained when a distribution of trap states, present in the D–A blend, is taken into account. Using impedance spectroscopy (IS), we detect mid-gap trap site distributions in several OPD devices. As predicted by the simulations, a systematic/controlled decrease of the trap concentration from ~$3.5 \times 10^{15}$ to ~$1.0 \times 10^{15}$ cm$^{-3}$ indeed results in a one order of magnitude decreased $J_D$. Noise measurements performed on a number of OPDs show a weak, if not absent, frequency dependence from 10 Hz onwards, and a proportionality to $\sqrt{J_D}$, highlighting the major role played by the dark current in OPDs. The discovery of the relations between mid-gap traps and $J_D$, reported in this paper, refocuses the current optimization routines, targeting material properties rather than device engineering. Moreover, $E_{CT}$ determines the thermal lower limit of $J_D$ to seek for and provides a metric for judging how far $J_D$ is from this fundamental limit.

## Results

**The role of dark current on detectivity.** The specific detectivity is proportional to the external photovoltaic quantum efficiency (EQE) and inversely proportional to $S_n$:

$$D^* = \frac{q\lambda\sqrt{A}}{hc}\frac{\text{EQE}}{S_n}. \tag{1}$$

With $q$ as the elementary charge, $\lambda$ the wavelength, $h$ the Planck constant, $c$ the speed of light, and $A$ the device area. The best organic systems show EQEs approaching 100%, narrowing the room for improvement by increasing EQE. Despite these high EQEs, $D^*$ is limited to around $10^{13}$ Jones[8,15–22], far below the background-limited infrared photodetection limit (BLIP limit), which assumes EQE of 100% and the background radiation as the only source of noise. This discrepancy is a consequence of the high $S_n$ observed in OPDs and represents the main limiting factor for this device class to approach the BLIP limit. In Fig. 1a, this issue is visualized for two of the systems studied in this work: the role of EQE and $S_n$ are compared for P4-Ph4-DIP:C$_{60}$ and ZnPc:C$_{60}$, which, besides having representative CT energies among the studied systems, better represent the state-of-the-art EQE of current diode-based OPDs. See Supplementary Table 2 for details about the materials. The BLIP limit is shown as a black dot-dashed line. Symbols indicate $D^*$ given in literature and they are typically many orders of magnitude below this limit. The real $D^*$ of the P4-Ph4-DIP:C$_{60}$ device (solid line) would improve only by one order of magnitude if an EQE maximum of 100% would be

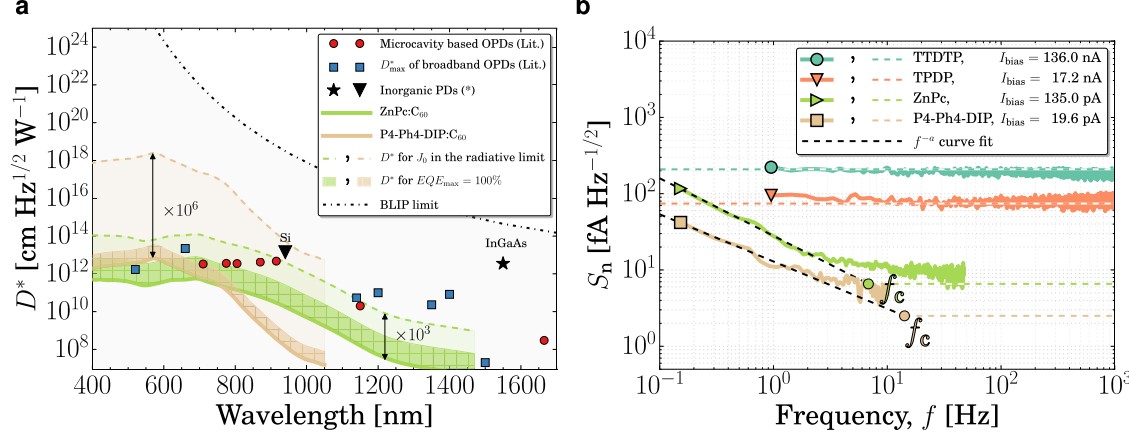

**Fig. 1 Specific detectivity and spectral noise density. a** $D^*$ of two donor:C$_{60}$ (6 mol%) material systems assuming shot noise at −1 V and EQE as measured (solid lines), shot noise at −1 V and normalized EQE at the maximum of the spectrum (hatched region), and shot noise in the radiative limit and EQE as measured (dashed lines). Symbols show data from literature[8,15–22]. **b** $S_n$ of four donor:C$_{60}$ (6 mol%) material systems. Dashed lines represent the shot noise calculated at $I_{bias}$. See Supplementary Figs. 15 and 16 for more results. (*) Hamamatsu K1713-05.

reached. On the other hand, if $S_n$ is assumed to be dominated by the shot noise and is calculated in the radiative limit, $D^*$ can be improved by six orders of magnitude, considering the real EQE. A similar behavior is also shown for ZnPc:C$_{60}$. From these examples, it is clear that there is large room for improvement, and noise current has to be significantly improved in order to achieve a higher detectivity.

The measured $S_n$ embraces several unknown sources. However, a comparison of $S_n$ to the shot noise contribution (colored dashed lines in Fig. 1b) reveals that, above the noise corner $f_c$, i.e., the frequency at which the noise assumes a $1/f$ dependence, the dark current represents the dominant noise source in OPDs. As shown in Fig. 1b, $S_n$ approaches the theoretical shot noise level, calculated as $\sqrt{2qI_{bias}}$, where $I_{bias}$ represents the current driven through the device at a bias voltage of $-0.8$ V. This highlights the importance of studying dark currents in OPDs. Because $S_n$ is mainly determined by $J_D$, suppressing $J_D$ translates directly into higher $D^*$.

**Diode saturation current generated via charge-transfer states.** In order to understand the origin of the dark current, we have reviewed the present perspective on how $J_D$ is generated. The dark current as well as the open-circuit voltage ($V_{OC}$) have been shown to relate to the energy difference between the highest occupied molecular orbital (HOMO) of the donor and the lowest unoccupied molecular orbital (LUMO) of the acceptor[23,24]. In fact, at open-circuit conditions, free charge carriers recombine through CT states, connecting $V_{OC}$ and $E_{CT}$[25]. While $E_{CT}$ is linked to the energy levels of donor and acceptor, a direct relation cannot be drawn as it hides polarization effects and binding energies, which can strongly modify the energy value, depending on the materials and mixing ratios[26].

In an ideal diode under illumination, the dark saturation current $J_0$ and $V_{OC}$ are linked by

$$J_0 \approx J_{SC} \exp\left(-\frac{qV_{OC}}{k_BT}\right), \text{ for } V_{OC} \gg \frac{k_BT}{q}. \qquad (2)$$

$J_{SC}$ is the short-circuit current of the photodiode under illumination, $k_B$ the Boltzmann constant and $T$ the absolute temperature. The following relation between $V_{OC}$ and $E_{CT}$ was derived on the basis of the detailed balance theory[27]. It has been successfully employed to explain the dependence of $V_{OC}$ on CT state properties, including the often observed correlation between $V_{OC}$ and $E_{CT}$, the dependence of $V_{OC}$ on D–A interface area[28]

and non-radiative recombination:[29]

$$V_{OC} \approx \frac{E_{CT}}{q} - \frac{k_BT}{q}\ln\left[\frac{2\pi q}{h^3c^2}\frac{(E_{CT}-\lambda_{CT})}{J_{SC}EQE_{EL}}f_{CT}\right], \qquad (3)$$

where $EQE_{EL}$ is the external quantum efficiency of electroluminescence, $\lambda_{CT}$ the reorganization energy of the CT state and $f_{CT}$ is proportional to the oscillator strength of the CT transition and the density of CT states in the blend[27].

While Eq. (3) successfully explains $V_{OC}$ in organic solar cells, it is not clear whether $J_0$, linked to $V_{OC}$ by Eq. (2), corresponds to the measured dark current at negative bias voltages ($J_D$). To investigate this, we fabricate a series of devices optimized in terms of dark currents (selective contacts, appropriate blocking layers and optimized device engineering), based on different donors blended with C$_{60}$ at 6 mol%. Here, low donor content bulk heterojunctions (BHJs) have been chosen to ensure a comparable morphology, which is known to depend on D–A mixing ratio[26,30], miscibility[31], and aggregation properties[32]. Details about the series of donors and materials used to fabricate the devices can be found in Supplementary Tables 2 and 3.

In Fig. 2a, the experimental dark $JV$ characteristics of seven different devices employing different donor molecules combined with C$_{60}$ are shown. In this series, $E_{CT}$ is increased from 0.85 eV (TTDTP:C$_{60}$) to 1.58 eV (P4-Ph4:DIP:C$_{60}$), as indicated in the legend. At reverse voltages, the dark current indeed decreases with increasing $E_{CT}$, however, not as predicted by Eq. (2), from which an exponential dependence is expected. For the blend with the highest $E_{CT}$ (P4-Ph4-DIP:C$_{60}$), dark currents as low as $10^{-7}$ mA cm$^{-2}$ at $-1$ V bias were achieved, which is among the lowest values reported for state-of-the-art OPDs[3]. To accomplish such a low dark current, several optimization strategies were performed, described in Supplementary Note 1. These approaches are also reflected in the remarkably low noise corner achieved in these devices, in the range of 0.3–150 Hz, c.f. see Fig. 1b and Supplementary Fig. 15, which are lower than that for recently reported high-performance OPDs[33,34].

For the sake of comparison, we kept the same structure for all devices shown in Fig. 2, see also the inset. This implies that for high-$E_{CT}$ combinations, from Spiro-MeO-TPD towards lower HOMOs, an injection barrier for holes under forward bias is formed between donor and electron blocking layer (EBL: MeO-TPD). Such a barrier only affects the forward regime of the $JV$ curves up to the space-charge-limited-current region. For

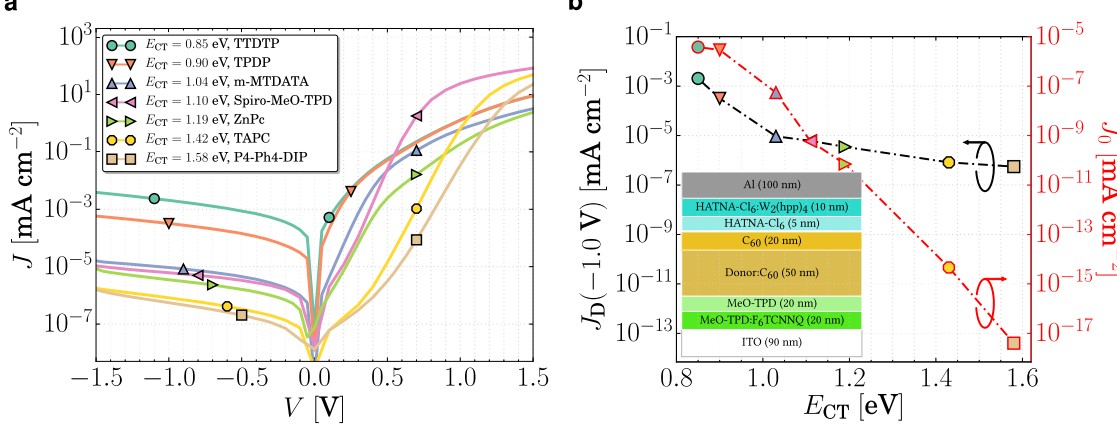

**Fig. 2 Experimental and ideal dark $JV$ characteristics versus $E_{CT}$. a** Experimental dark $JV$ characteristics of different blends with different $E_{CT}$. **b** $J_D$ extracted from (**a**) at –1 V (black left y-axis) and ideal $J_0$, calculated through Eqs. (2) and (3) (red right y-axis). For the calculation of $J_0$, we extracted $E_{CT}$, $\lambda_{CT}$, and $f_{CT}$ from sensitively measured EQE spectra by fitting the CT state feature with a Gaussian function, and $EQE_{EL}$ was estimated from the non-radiative voltage losses, both as described previously[14] (see Supplementary Table 4 and Supplementary Fig. 9). Inset shows the device stack used. Legend from (**a**) is also valid in (**b**). Dash-dotted lines in (**b**) are guides to the eye.

low-$E_{CT}$ combinations, having TTDTP, TPDP, or m-MTDATA as donor, an extraction barrier under reverse bias is formed at the same interface. This barrier can only decrease the reverse $J_D$, as holes are prevented from being extracted. However, even for these systems, the same trend of $J_D$ versus $E_{CT}$ is observed. In the absence of extraction barriers, this relation between $E_{CT}$ and $J_D$ would only be strengthened. The extraction barrier also affects the forward region of the $JV$ curves, which is reflected in Fig. 2 and will be discussed later in more detail.

Despite these low dark currents and the observed scaling of $J_D(-1\,V)$ with $E_{CT}$, there are two main issues that need to be pointed out: (i) $J_D$ at reverse voltages of all OPDs is at least three orders of magnitude higher than the ideal diode dark saturation current (see Fig. 2b) and (ii) Eqs. (2) and (3) do not account for any field dependence of the dark current at reverse bias. However, the experimental data presented in Fig. 2a clearly show an increasing dark current upon increasing the absolute reverse voltage. In the following sections, we address fundamental characteristics of organic diodes based on the fullerene $C_{60}$ that help us to clarify the two aforementioned points and understand why the experimentally measured $J_D$ deviates from $J_0$ calculated via Eq. (2).

**Traps are the main source of reverse dark current in OPDs**. Trap states with intra-gap energies are commonly observed in organic materials and devices due to their disordered nature[35], structural defects and the presence of impurities[36]. Several publications address the limitations on charge transport[36–38], increase of recombination rates[39] and change of recombination dynamics[40] caused by these states. However, only very few studies investigated the influence of trap states on $J_D$[41]. Drift-diffusion models with band-tail[42] and mid-gap[43] trap states were employed to reproduce the experimental $JV$ characteristics of organic solar cells and OPDs, respectively. However, the trap density of states was not characterized and thus it is unclear whether the number of traps assumed corresponds to that of the real device. Moreover, the field dependence was either ignored or described by an

electronic band structure model, making its application for organic materials questionable. A consistent experimental observation of the impact of traps on $J_D$, supported by a theoretical modeling, is still missing.

The microscopic properties of a D–A system are related to the electronic characteristics of the device, e.g., donor concentration usually affects $V_{OC}$, because the number of CT states and their energy change. This gives us also insight into trap states, which could likewise arise from the D–A interaction and depend, therefore, on the D–A mixing ratio. If a correlation between mixing ratio and concentration of traps exists, also the impact on $J_D$ can be investigated. From Fig. 2, we know that $E_{CT}$ strongly influences $J_D$, which means that a D–A system and a range of mixing ratio has to be found in which $E_{CT}$ is constant. A careful analysis of the material systems shown in Fig. 2 revealed that in the TPDP:$C_{60}$ system, from 6 mol% to around 27 mol%, this condition is fulfilled.

To enlighten the relation among mixing ratios, trap and $J_D$, we fabricated devices comprising different concentrations of TPDP:$C_{60}$ using the same device architecture as shown in the inset in Fig. 2b. As the concentration of TPDP in $C_{60}$ decreases from 26.7 to 6.0 mol%, $J_D$ also decreases by approximately one order of magnitude, as shown in Fig. 3a. At the same time, $E_{CT}$ increases less than 20 meV (see Supplementary Table 5 and Supplementary Fig. 10), which, according to Eqs. (2) and (3), cannot explain the significant decrease of $J_D$. In addition to the improved $J_D$, this result raises the question whether a correlation with the amount of traps can be found.

Measuring traps in organic solids is a rather difficult task due to the lack of simple techniques to access their concentration and energetic characteristics. Nonetheless, a widely applied method was proposed by Walter et al. based on the capacitive response of these states, whose occupation varies with the modulation of the signal[44]. On the one hand, this enables the characterization of defects by a straightforward technique, such as IS. On the other hand, such a technique probes many different effects in the

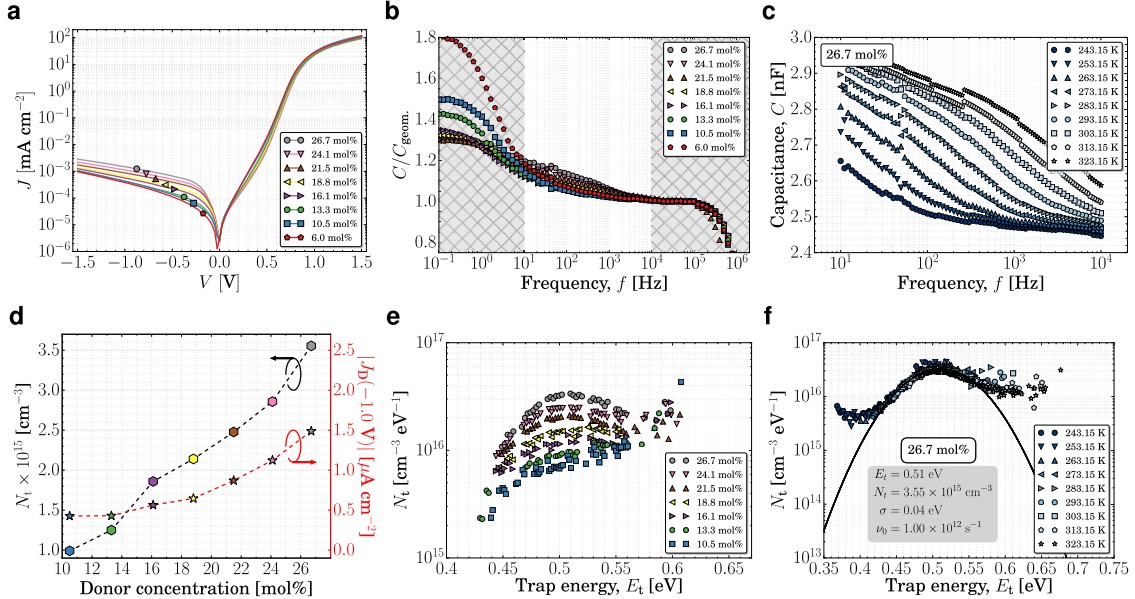

**Fig. 3 Increasing the trap density in TPDP:$C_{60}$ devices. a** Dark $JV$ curves, **b** normalized capacitance, and **e** $N_t$ at 293.15 K of TPDP:$C_{60}$ blends with different D–A mixing ratios. Due to the small capacitive contribution of the traps at very low concentrations, the method fails in reconstructing the trap density of states. Therefore, the results of the 6.0 mol% device are not shown. **c** Capacitance and **f** $N_t$ for 26.7 mol% at different temperatures reconstructed as proposed previously[44]. **d** Trap density (left black y-axis) and $|J_D(-1.0\,V)|$ (right red y-axis) versus donor concentration. Trap energies represents the energy difference from $C_{60}$ LUMO (LUMO$_{C_{60}}$ - $E_t$)[62]. The hatched areas in (**b**) show the frequency ranges excluded from the trap analyses as discussed in the text. $C_{geom.}$ refers here to the capacitance at 100 kHz, where the geometric capacitance dominates.

device, especially in an organic one, which can show identical capacitive spectra to those generated by traps, inspiring discussion in the community whether such a measurement can in fact reveal the trap characteristics[45–47]. Special caution should be taken when using this method in low-mobility materials and in devices where energetic transport barriers are present. As both are the case in our devices, in Supplementary Note 3 we discuss the limitations of the method and show that for our devices, IS can show meaningful results.

Excluding the range below 10 Hz, where the capacitance spectra are dominated by resistance of the layers, which, as expected, increases for lower concentrations; and above 10 kHz, where the geometric capacitance and the series resistance of the contacts controls the spectra, Fig. 3b shows an increase in the capacitance, which we attribute to traps. By studying this region in more detail at different temperatures, as depicted in Fig. 3c for 26.7 mol%, and overlapping the data by the appropriated choice of $\nu_0$, we can reconstruct the trap density plot, as shown in Fig. 3f for this concentration and in Fig. 3e for one temperature per concentration. The reconstructed trap densities for every concentration have been fitted with a single Gaussian distribution function each, similar to Fig. 3f, from which $N_t$, $E_t$, and the broadness of the Gaussian, σ, were extracted (see inset, Supplementary Note 2, and Supplementary Figs. 11 and 12). $N_t$ is summarized in Fig. 3d and compared to the absolute value of $J_D(-1.0\,V)$. The increase of $J_D$ with increasing $N_t$ suggests that extra-dark current is produced by the generation of charges via trap states.

As the TPDP concentration decreases in the blend, the amount of trap states becomes lower, as shown in Fig. 3e. For concentrations below 10.5 mol%, the sensitivity limit of the measurement is reached and the method fails in reconstructing the density of trap states for this material system. Nevertheless, the presence of traps cannot be excluded.

While from Fig. 3 it is clear that traps are connected to $J_D$, we can also model the thermal generations of charges contributing to $J_D$ via these traps to reveal their role on $D^*$. This can be done in the framework of the Shockley-Read-Hall (SRH) theory[48]. In a p-i-n diode, the generation rate in reverse bias can be written as:

$$G = \frac{\beta_{SRH} N_t n_i}{2\cosh\left(\frac{E_t - E_i}{k_B T}\right)}. \tag{4}$$

Here $\beta_{SRH}$, $n_i$, and $E_i$ are the recombination rate, the intrinsic charge carrier concentration, and the mid-gap energy, respectively[12]. Equation 4 describes, as observed experimentally, that the generation increases linearly with $N_t$ and is more efficient when $E_t = E_i$, i.e., mid-gap traps are more relevant for $J_D$. This is a consequence of the trapping/detrapping probability of a charge in a two-step process: an electron is firstly excited from the valence band to the trap state, followed by a second excitation, which releases it to the conduction band. A charge contributes to $J_D$ only after a double thermal excitation and the final rate depends on both activation energies. As schematically shown in Supplementary Fig. 7, if $r_2$ increases as $E_t$ approaches the conduction band, $r_4$ decreases concomitantly. In fact, the highest generation rate is achieved around $E_i$, decreasing exponentially towards valence and conduction band (see Supplementary Note 4 for more details).

The experimental reverse current densities are generally not constant, but increase with the magnitude of the reverse voltage. This behavior is not reproduced by the simple models of SRH generation or thermal generation of the effective gap, which do not contain a field dependence. However, in thin-film devices, i.e., vertical dimensions, at −1 V, electric fields of around $10^5$ V cm$^{-1}$ can be achieved. In this electric field regime, it is reasonable to assume that, due to energy level bending, the trap energy depth is lowered by the applied electric field.

The Poole-Frenkel model describes field-dependent generation by stating that, if a carrier is bound in a trap state, the energy landscape can be bent by an external field such that the effective energy necessary for escaping the trap is diminished. For a trap of zero-field depth $E_t$, an approximation for this effective depth $E_{t,eff}$ is[49]

$$E_{t,eff} = E_t - \left(\frac{q^3 F}{\pi \epsilon \epsilon_0}\right)^{\frac{1}{2}}, \tag{5}$$

with the absolute value of the electric field $F$ and the relative and vacuum permittivity, $\epsilon$ and $\epsilon_0$, respectively. By implementing the modified trap depth for calculating the SRH efficiency in a single-layer drift-diffusion simulation with ohmic contacts, where $F$ is calculated locally, taking the internal charge distributions into account, we study the contribution of the traps on $J_D$ for this D–A system. Fig. 4a shows these simulations and compares them with the experimental data. See Supplementary Table 6 for parameters used in the simulations. Using the trap parameters extracted from the trap distribution as input in the simulation, the model describes the increase in $J_D$ upon increase of the amount of traps, achieving a good agreement in the voltage dependence and magnitude of the dark current. As discussed by Fallahpour et al., a generic trap distribution at mid-gap is also able to reproduce the magnitude of $J_D$ observed in many OPDs[43]. However, besides failing in reproducing the field dependence, such approach is debatable, given the major influence of $N_t$ and $E_t$ on $J_D$. The impacts of σ and $E_t$ are discussed in Supplementary Fig. 22.

At low forward voltages, the simulated current densities are higher than the experimental values, which is a consequence of the extraction barrier for holes in these devices (see the schematic energy diagram in Fig. 4c). This barrier arises from the difference of HOMO level of the donor TPDP[15] and the EBL, MeO-TPD[50]. The effect vanishes when the EBL thickness is decreased to 5 nm, as shown in Fig. 4a, indicating that holes can be extracted, possibly because the underneath layer is not completely covered. However, this also affects the blocking property of the layer, which ultimately precludes the study of $J_D$ versus trap concentration, as a larger amount of electrons is injected under reverse bias. Besides causing S-shapes on $JV$ curves of organic solar cells[51], the extraction barrier modifies the onset of the forward current[52]. The voltage at which current starts to flow depends on the built-in field, which is determined by MeO-TPD in this device. Moreover, as thickness of the EBL increases, the electric field becomes weaker for thicker EBLs, which reduces the current (see Supplementary Fig. 18).

The extraction limitation in TPDP:C$_{60}$ devices also becomes evident when simulating a barrier-free device, such as the well-studied ZnPc:C$_{60}$ (50 wt%) system, shown in Fig. 4b. The model (red dashed line) is able to describe entire experimental dark $JV$ curve with a good agreement (see Supplementary Fig. 17 for the trap characterization of this device). Since only the D–A systems were changed, it explains the deviation observed in Fig. 4a. The extraction barrier affects also the reverse region, mainly in the low-field regime as shown in the inset of Fig. 4a, where the experimental current density is lower than the simulated one, due to the extraction barrier faced by holes. See Supplementary Fig. 18 for more details.

It is common to associate the ideality factor, i.e., the slope of the exponential region of the $JV$ curve ($n_{id}$) to recombination processes, and it should approach two when trap-assisted recombination dominates. From a first analysis, one could conclude that as the trap concentration increases, in Fig. 3a, $n_{id}$ decreases. However, because of the extraction barrier, the information given by $n_{id}$ is misleading and its analysis non-trivial. In fact, by locally accessing $n_{id}$ at different temperatures, we can define a more meaningful region where $n_{id}$ can be characterized[53]. As shown in Supplementary Fig. 20, $n_{id}$ is

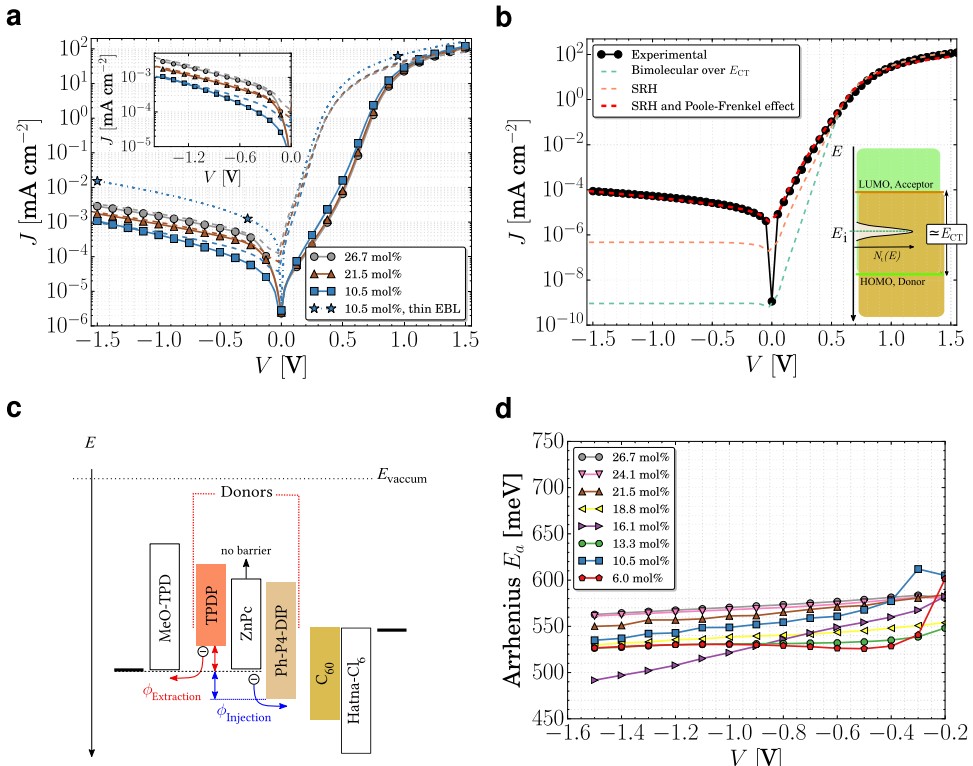

**Fig. 4 Simulated *JV* characteristics of different D–A systems.** Simulated (dashed lines) compared to experimental (symbols and solid lines) for devices based on: **a** TPDP:$C_{60}$ at three representative concentrations and **b** ZnPc:$C_{60}$ (50 wt%). Also shown in (**b**) are the *JV* characteristics for SRH process in the absence of Poole-Frenkel effect and bimolecular generation over $E_{CT}$. Inset in (**a**) shows the reverse region in detail and in (**b**) a schematic representation of a mid-gap trap distribution in the donor–acceptor system. Mobilities and recombination rates were optimized to achieve a good agreement with the experimental data (see Supplementary Table 6). **c** Schematic representation of the energy level of the three systems showing an extraction and an injection barrier. A barrier-free system is also shown. **d** Activation energy for TPDP:$C_{60}$ extracted from temperature-dependent *JV* measurements. The current was measured at temperatures varying from 223.15 to 303.15 K with $\Delta T = 10$ K. For each bias, the logarithm of the current was plotted versus $1/T$, and $E_a$ was extracted from the slope of the curve. The fit for $V = -1.5$ V is shown in Supplementary Fig. 21.

between 1.7 and 1.8 for all devices, indicating that trap-assisted recombination is the dominating process. However, no clear trend with the amount of traps could be observed, which can still be a result of the extraction barrier, whose effect seems to get more pronounced for high-concentration devices. This observation is in agreement with Supplementary Fig. 10, where $E_{CT}$ slightly decreases with the TPDP concentration, indicating that the extraction barrier increases.

At higher voltages, the *JV* curves eventually reach the space-charge-limited-current regime, where the current density is primarily determined by the carrier mobility and the device thickness[49,54]. In this regime, our simulations match the experiments for both the ZnPc and the TPDP device, confirming that the mobilities assumed for the simulations are adequate.

The importance of the energy level bending and the traps can be visualized in Fig. 4b. In the absence of traps, only charge carriers thermally excited over the effective gap, i.e., $E_{CT}$, contribute to current in reverse bias. This leads to values of around $10^{-9}$ mA cm$^{-2}$, similar to the $J_0$ estimated from the quantum efficiency measurements of this device (see Supplementary Table 4). By including SRH generation through the measured trap states, the simulated $J_D$ increases almost three orders of magnitude, however, still not fully reproducing the experimental data, as both magnitude and field dependence are not reached solely by the SRH generation. Ultimately, the experimental data can be reproduced by including the Poole-Frenkel effect.

The simulations and the experimental data suggests that activation energy for charge detrapping is field-dependent (see also Supplementary Fig. 19). This energy $E_a$ can also be accessed via

temperature-dependent measurement of the dark current. In Fig. 4c, the Arrhenius analysis of $E_a$ for TPDP:$C_{60}$ at different voltages reveals that its magnitude decreases when higher fields are applied to the device. This is a direct consequence of the Poole-Frenkel effect: the higher the applied voltage, the lower the barrier and, therefore, the lower the activation energy. Note also that $E_a$ is very similar to $E_t$ measured via IS, which is consistent with the model developed here.

A quantitative agreement of the absolute $E_a$ values is however not possible, since the experimental values are affected by other thermally activated processes that we did not take into account in the simulations, such as charge transport and overcoming energy barriers. We speculate that charge transport is also responsible for the decrease of the activation energy in low-bias regime for low-concentration devices.

**The interplay between CT states and trap states.** Besides quantitatively describing the dark current of ZnPc and TPDP donors, by a variation of $E_{CT}$, our model is also able to reproduce the entire range of experimental $J_D$, depicted in Fig. 2a. This variation is shown in Fig. 5, where $E_{CT}$ is increased from 0.9 to 1.3 eV. Here, we targeted the effect of $E_{CT}$ versus traps, therefore all parameters were kept constant, except for $E_{CT}$, explaining the deviation of the absolute value of $J_D$ versus $E_{CT}$. Nonetheless, it is clear that, with the inclusion of traps in the model, the experimental $J_D$ can be reproduced. To support this finding, the trap concentration of all devices was characterized. For all of them, $N_t$ ranges from $10^{15}$ to $10^{16}$ cm$^{-3}$, obtained from the fit of the

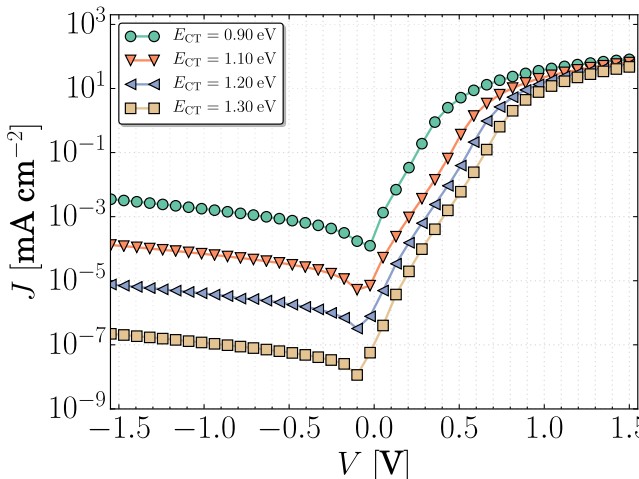

**Fig. 5 Simulated dark *JV* characteristics for different $E_{CT}$.** $N_t$, $E_t$, and $\beta_{SRH}$ were kept constant as described in Supplementary Table 6.

measured spectra to a Gaussian distribution, indicating a rather general trend for donor:$C_{60}$ BHJs (see Supplementary Figs. 13 and 14). While many reports describe an exponential density of trap states, for the materials investigated within this work and the energy range probed by IS, the measured spectra resemble a Gaussian distribution (Fig. 3f), which is in agreement with previous investigations in organic systems based on different preparation techniques[55–58]. Given that traps have been observed not only in small molecule fullerene-based BHJs, but also in many polymer-based D–A structures[38,59,60] and non-fullerene-based devices[58], it is reasonable to assume trap-assisted dark current generation is an important source of non-idealities in the experimental dark *JV* curves reported in literature. The final dark current is dominated by trap-assisted generation, which is determined by the trap properties, as shown in Supplementary Fig. 22. These findings highlight that, in addition to optimization routines to reduce $J_D$ from a device engineering perspective, future research should also focus on understanding the origin of such states and parameters governing trap creation.

We also want to point out that the trap states characterized above are not observed in sensitive optical measurements of the photocurrent or EQE spectra, as previously reported[61]. The sensitivity EQE spectra reveals only characteristic absorption peak attributed to CT states (see Supplementary Fig. 9). These states are accounted for when calculating $J_0$, as described above.

## Discussion

In this paper, we develop a comprehensive model, which can quantitatively explain the reverse currents in BHJ photodiodes. $J_D$ follows the same trend with $E_{CT}$ as its theoretical value $J_0$, but is orders of magnitude higher. This can be explained by trap-assisted generation in a field-dependent version of the SRH model. By characterizing the trap distribution of different materials and blend concentrations, we show that $J_D$ scales with the total trap density and forms the main generation path in the studied OPDs. The commonly observed voltage dependency can be understood as the enhancement of emission rates assisted by the energy-barrier lowering caused by the reverse applied bias. By using different approaches, we reduce $J_D$ to around $10^{-7}$ mA cm$^{-2}$ and demonstrate that shot noise dominates the noise current. The shot-like behavior can be attributed to the detrapping of charge carriers, where an energy barrier of $E_t$ must be overcome for the generation.

Although our results point to an interfacial interaction between donor and acceptor, the origin of trap states is still unknown and further research needs to be done to clarify this aspect. Detectivity

is limited by high noise currents, especially in the NIR regime. Understanding the molecular and morphological parameters ruling the formation of trap states is essential to pave a path towards its suppression and, therefore, a major increase in detectivity.

## Methods

**Device preparation.** Organic layers used in the devices were thermally evaporated on glass substrates covered by pre-structured ITO contact (thin-film devices) at ultrahigh vacuum (pressure <$10^{-7}$ mbar). Before use, the organic materials were purified 2–3 times via sublimation. The overlap of the bottom and top contact (Al, 100 nm, Kurt J. Lesker) defines the device active area (6.44 mm²).

**JV characteristics.** Illuminated *JV* characteristics were measured using a source measurement unit (Keithley SMU 2400). The devices were illuminated with a spectrally mismatch-corrected intensity of 100 mW cm$^{-2}$ (AM 1.5 G) provided by a sun simulator (Solarlight Company Inc., USA). The intensity is controlled by a Hamamatsu S1337 silicon photodiode. Dark *JV* characteristics were measured with a high-resolution SMU (Keithley SMU 2635). Every measurement data point was acquired after steady-state conditions were achieved. The measurement is controlled by the software SweepMe! (https://sweep-me.net/).

**External quantum efficiency (EQE).** The current generated by the device under monochromatic light (Oriel Xe Arc-Lamp Apex Illuminator combined with Cornerstone 260 1/4 m monochromator (Newport, USA)) is measured with a lock-in amplifier (Signal Recovery SR 7265). A mask (2.78 mm²) is used to avoid edge effects. The same procedure is followed to monitor the light intensity, measured with a calibrated silicon diode (Hamamatsu S1337 calibrated by Fraunhofer ISE). EQE is obtained by the ratio of charge carriers generated by the device with the number of incoming photons.

**Sensitive external quantum efficiency (sEQE).** A chopped monochromatic light (140 Hz, quartz halogen lamp (50 W) used with a Newport Cornerstone 260 1/4 m monochromator) is shined to the device. The current generated at short-circuit conditions is amplified before being measured by a lock-in amplifier (Signal Recovery 7280 DSP). The time constant of the lock-in amplifier was chosen to be 500 ms and the amplification of the pre-amplifier was increased to resolve low photocurrents. Light intensity was obtained by using a calibrated silicon (Si) and indium-gallium-arsenide (InGaAs) photodiode.

**Impedance spectroscopy.** Capacitance-frequency spectra were measured by an Autolab PGSTAT302N at 0 V bias using a sinusoidal signal with 20 mV of amplitude, from 10 KHz to 1 Hz. The sample was measured at different temperatures in the sample setup described above for *JV*. Because the reconstructed density of traps is proportional to d$C$/d$\omega$, minor noise effects in the measurement leads to a considerable scattering in the density of traps. For better data visualization, relative deviations higher than 30% were excluded. More details can be found in Supplementary Note 2.

**Noise measurements.** An input stage transimpedance amplifier is used to convert the current through the OPD into the voltage $v_{o1}$, followed by two stages of high-pass filters (HPF) and two stages of signal amplification (gain) plus low-pass filtering (LPF). The output signal $v_{out}$ is then sampled at 4–12 million points in real-time using Rohde&Schwarz RTO 2044 oscilloscope with 16-bit of resolution. The spectrum of $v_{out}$ is calculated using the Welch's power spectral density (PSD) estimate method in MATLAB. The LPFs and HPFs significantly attenuate the signal power content outside of the target frequency bandwidth. This prevents any mistranslation of nontarget signal power into the target measurement bandwidth. More details can be found in Supplementary Note 5.

## Data availability

The data that support the findings of this study are available in Materialscloud with the identifier doi:10.24435/materialscloud:sq-wv

## Code availability

All code used to generate the data that support the findings of this study are available from the corresponding authors upon reasonable request.

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

## Acknowledgements

This work was supported by the German Federal Ministry for Education and Research (BMBF) and by the German Research Foundation (DFG) within the Cluster of Excellence Center for Advancing Electronics Dresden (cfaed) and the DFG projects HEFOS (Grant No. FI 2449/1-1) and Photogen (Grant No. VA 1035/5-1). B.K.B. acknowledges funding from DFG Priority Programme FFlexCom under the project FlexARTwo (LE 747/52-2 and EL 506/22-2). J.K. acknowledges funding by the Deutsche Akademische Austausch Dienst (DAAD). J.B. thanks for the financing from Sächsische Aufbaubank through project InfraKart (Grant No. 100325708). Furthermore, we acknowledge N. Sergeeva, Prof. Dr. T. Kirchartz, and Prof. Dr. B. E. Pieters for fruitful discussions.

## Author contributions

J.K., D.S., H.K., K.V., and K.L. designed the experiments, prepared the photodetectors and optimized the devices for dark current. J.K. performed the standard characterization of detector and measured the trap density. A.H. performed the drift-diffusion simulations. J.K. and J.B. measured sensitive EQE spectra and V.C.N. calculated the voltage losses of the devices. C.K. synthetized TTDTP. B.K.B. developed the noise measurement setup, performed the noise measurements, and F.E. supervised this work. J.B., D.S., A.F., H.K., and K.V. supervised their team members involved in the project, K.V. and K.L. supervised the overall project. All authors contributed to analysis and writing.

## Funding

## Competing interests

Dr. Axel Fischer is co-founder of "Axel Fischer und Felix Kaschura GbR" which provided the measurement software "SweepMe!" (sweep-me.net). The name of the program is given in the manuscript. The other authors declare no competing interests.
