## [Peer Review File. · Nature Communications]

Reviewer #1 (Remarks to the Author):

This communication reports on the origin of dark reverse saturation current of OPDs based on systematic analyses. Supplementary Note 1-3 show the author's effort to objectively explain the cause of dark current. After carefully reading this communication, I feel the suggested correlation between the effective charge-transfer state energy and measured dark current is very reasonable. I strongly recommend publication of this manuscript. Nonetheless, the following points should be addressed before the final acceptance.

1. From the author's findings on J_{D} vs E_{CT} and E_{CT} vs trap states, it seems reasonable to assume that larger E_{CT} can lead to higher density of trap states. If so, what would be clear explanation on this from material side?
2. Though impedance analyses fitted with Gaussian function were quite successful on analyzing trap distribution, there have been many reports that showed success of exponential distribution. The authors need to explain why they chose Gaussian. (PRB 84, 195209, 2011)
3. If the origin of dark current is thermal release of trapped carriers as the authors argued, I guess shallower traps would have more dominant contribution to dark current than the case of deep traps. It would be helpful if the authors explain this point.
4. The authors introduced Poole-Frenkel model to explain field dependence of the measured dark current. In this case, the assumption of even distribution of electric field along with the thickness direction of the active layer is required. However, in this report, three different layers of p, i and n with different resistances are located along with the thickness direction. The authors need to give reasonable explanation on this issue to use equation 5.
5. Some minor points
 - Page 2, line 52 : The authors argued that the ideal dark saturation current is thermally activated. However, the reverse saturation current of conventional PN junction diode is limited by both diffusion current and thermal generation. Please give reasonable modification on this. (S. M. Sze and Kwok K. NG, Physics of Semiconductor Devices)
 - Page 5, line 134 : The authors mentioned that low donor content BHJs have been chosen to ensure minimum morphological effects. Relevant references are required here for researchers who are not familiar with molecular semiconductors.
 - It would be better to explain why the authors limited the sweep range of J-V characteristics to -1.5V - 1.5 V. Note that CMOS image sensors require much higher sweep range to -5V.

Reviewer #2 (Remarks to the Author):

In this paper Kublitski and co-workers report a quantitative model describing the role of mid-gap trap states as the key source of noise in organic photodiodes. This is important because, as a result

of these traps, organic photodiodes show high noise spectral density (S_n), which limits their specific detectivity to around 10^{13} Jones. Although the physics of these interfacial trap states in donor-acceptor organic bulk-heterojunctions have been well-characterized in the past for solar cells, their role in limiting the photodetection performance at reverse bias operation is not fully understood. The comprehensive understanding provided by this study will promote the development of organic photodiodes with high detectivity levels, although the origin of these traps remains unknown. This manuscript is highly technical and will be of particular interest to researchers specialized on organic electronics and photodetector research. It is overall well written, but the clarity can be improved especially for non-specialists. Below are some additional comments:

- I am confused about the description of Fig. 4a (Line 253-265). The poor fit between simulated current and experimental data in the forward bias region is attributed to the thickness of the EBL. In that case is it possible to get a better fit using a thicker EBL in the simulation or does that lead to poor fitting at reverse bias? Simulated results for various EBL thickness should be provided and compared with experimental data in Fig 4a and Fig. S14.

- Figure 4b: does the simulated current (red and orange dashed lines) reproduce the dip in current at short-circuit?

- Some mistakes found in the reference list: e.g. Ref. 30 and Ref. 50.

Reviewer #3 (Remarks to the Author):

In a combined experimental and modeling study Kublitski et al. show that electronic traps at the donor-acceptor interfaces (so in the bulk) affect the shot noise and dark current in organic photodetectors. Though this idea is not novel, they show for the first time experimentally (using impedance measurements) that a great variety of donor-acceptor blends have low trap densities (in the range of 10^{15} - 10^{16} cm⁻³) and that these trap states are very likely associated with the (volumetric) heterojunction formed between donor and acceptor material. The microscopic origin of the traps are not discussed other than in general terms of 'defects'.

Studying dark current in these devices implies a lot of control experiments. Most of these control experiments, albeit not all, have been performed. Doing this for a dazzling amount of material systems means even more work. Especially the experiment in which the TPDP-to-C60 ratio was systematically varied, is elegant, and the observed relationship between N_t and dark current insightful. My compliments!

The vast magnitude of material systems studied and large number of experimental techniques used, however, also makes the manuscript difficult to read (and write). Not all data can be shown but I sometimes fail to understand the selection made by the authors.

The discussion and interpretation of the impedance results should be improved. Using the fitting parameters of the trap states J_D is calculated using a relatively simple drift-diffusion model. Given my comments below on the experimental accuracy of the fitting, more simulation data can help to get better physical insights. Showing that the simulations and experimental results are consistent is a rather meagre conclusion, I find. Authors should perform a more thorough analysis on what the results tell us, what can be properly explained and what not.

Fundamental points

Measuring low density of trap states is very difficult, so the interested reader should be able to follow the argumentation and analysis of the authors step-by step.

1. The analysis of thermal admittance spectroscopy to derive the properties of trap states in low-mobility semiconductor devices is debated in literature. See for instance: H.S. Pang et al. Capacitive methodology for investigating defect states in energy gap of organic semiconductor, *Organic Electronics* 2018; F. Werner et al. Can we see defects in capacitance measurements of thin-film solar cells; S. Wang et al. Understanding thermal admittance spectroscopy in low-mobility semiconductors, *J Phys Chem* 2018. This paper should include a proper discussion on the use of impedance spectroscopy, incl. its limitations.

2. The authors should provide the 'raw' impedance spectroscopy data (of TPCP in Figure 3, and of the other donors in the Supplementary Information) and discuss measurement reproducibility

1. The authors should clearly show that it is possible to distinguish between deep traps and capacitive contributions (from for instance transport layers or additional layers in the device). I miss here a simple control experiment: does N_t (and J_D for a given electric field) stay constant with increasing thicknesses of the BHJ? Can the authors comment on this? Is it experimentally difficult?

3. The authors should elaborate on their choice of attempt-to-escape frequency as it directly relates to trap energies. Why do the authors use different values for this parameter for different donors?

4. The extracted trap densities are fitted with a single Gaussian but the quality of the fit is rather poor in most cases (Figure 10 in SI). The authors fitted trap distribution in the donor:C60 with a single Gaussian distribution that has its maximum at 0.51-0.57 (Figure 3b and SI Figure 10) and an energetic disorder of 0.04-0.10 eV. Looking at the data of ZnPc and Spiro-MeO-TPD, this choice is debatable, and thus not evidently points to the conclusion stated by the authors that 'the distribution of traps, ..., indicates a rather general trend for donor:C60 BHJs (p.1, line 306)'. Authors should elaborate on the fitting procedure and the extracted parameters so the reader can assess if the conclusion is inevitable.

The extracted trap parameters serve as input for the simulations.

1. Even in the case of the best fit (TPDP, Figure 3b), fitting the experimental data with a single Gaussian with maximum at around 0.5 eV (roughly midgap) with a sigma of 0.04 eV, implies that traps with energies lower than 0.4 eV and higher than 0.55 eV can be neglected. The authors should show quantitatively ('illustrate' with simulations) how important the exact distribution of traps is in the J_d - V simulations and thus the applicability of the model (SRH plus Poole Frenkel). It would be very convincing when the authors can show that the fit parameters of the trap distribution can simulate the absolute magnitude of J_D , its field dependence and temperature dependence simultaneously and consistently, at least for one donor:C60 system. If this is not possible, then a discussion on parameter sensitivity is required. In the current manuscript three fitting parameters are used to describe the trap distribution. Would a constant DOS not explain the J_D data (also considering that midgap states are most prolific, as seen from eq. 4)? Or what about a fit with a single Gaussian which has its energy maximum fixed at half the value of the CT energy, and N_t and sigma are the only two fit parameters? The authors are invited to simulate the alternative approaches, discuss the results and relate back to previous work of for instance Fallahpour (ref. 39 in the manuscript).

2. The authors should explain if (and why) they think V_{bi} should be taken into account when simulating the current-voltage data.

3. In this work, activation energies of J_D are determined using Arrhenius-behaviour, using a relatively small temperature range. The validity of this analysis is questionable. In many organic

semiconductors a temperature dependence of charge transport $\sim \exp(-a/T^2)$ rather than the expected thermally activated transport $\exp(-b/T)$ is observed. Authors should discuss the meaning and implications of this.

4. Extraction barrier can influence JD (SI Figure 14). Is this also relevant for the data shown in Figure 2, where most of the diodes seem to suffer from extraction barriers, judging from the forward currents. Please discuss possible implications (quantitatively).

Suggestions for further improvements

5. Figure 1a illustrates that experimental values of specific detectivity are much larger than the intrinsic values, as found in this paper as well as previous papers. The results of two of the 6 donor materials studied in this work are mentioned explicitly. Figure 1b shows the noise measurements of 4 of the 6 donor materials. To me these choices look random, also when considering that in the remainder of the text most work is done on the TPDP:C60 system. I suggest the authors to elude on their choices, include data of all 6 donor:C60, or be more consistent in another way (and perhaps take out the literature data in Figure 1a, it does not add much more than already mentioned in the text.) I could not find the original references to the D^* values of the inorganic PDs in Figure 1a.

6. Rather than the 2008 paper of Ramuz, include the more recent paper of Biele et al. (Adv. Mater. Technol. 2019, 4, 1800158) or alternative reports that reflects today's State of Art much better.

7. Introduce the meaning of I_{bias} when discussing the noise measurements.

8. Please look carefully at the colors of the different J-V curves in Figure 2a. I find it difficult to see what forward currents correspond to what reverse currents. Please discuss why forward current density is low in some of these devices.

9. Authors mention two times (on p.5 and p.6) that the observed scaling of JD with ECT is in agreement with equation 2. It is not! Equation 2 predicts an exponential dependence. This is different from the JD-ECT behavior plotted in Figure 2b.

10. Ref 23 and 24 deal with the relationship between V_{oc} and energy difference between HOMO of the donor and LUMO of the acceptor, but do not discuss it in the framework of dark current and photodetectors. A recent paper of Simone et al. (On the Origin of Dark Current in Organic Photodiodes, Adv Optical Mat 2019) does, so might be worth to include that one instead of 23 and 24.

11. Would be good to include a reference to a recent paper of Zhang et al. Sequentially deposited versus conventional nonfullerene organic solar cells, Adv Energy Materials, as this work also uses impedance spectroscopy to analyse interface trap states in BHJ, and find trap states centered between 0.5-0.6eV of 10^{17} cm^{-3} .

12. The 'dark' ideality factor of the J-V curves in Figure 3a decreases with increasing donor content in the film. This ideality factor is often associated with traps and recombination processes but this is not done in the present manuscript. Can the measured trap states explain the evolution of the dark ideality factor quantitatively?

13. Simple Poole-Frenkel predicts a \sqrt{E} dependence. This is frequently observed in organic devices. Please indicate if JD scales with applied electric field in these devices? If not, elaborate on this.

14. Authors promise us new optimization pathways to reduce noise current (p.3 line 71) but don't come back with clear recommendations. Please add, or take out this part in the introduction.

15. Supplementary p3. L 70 'It states that ' this indicates that, in the case of TPDP, the intrinsic processes happening in the blend dominate JD and the minor effect of selectivity is no longer observed. Following formal logic this first conclusion cannot be drawn from the sole observation that 'JD does not reduce by using p-HATNA-Cl6 in case of TPDP'. Please rephrase.

Rebuttal for:

Reverse Dark Current in Organic Photodetectors: The Major Role of Traps as Source of Noise

Jonas Kublitski, Andreas Hofacker, Bahman K. Boroujeni, Johannes Benduhn, Vasileios C. Nikolis, Christina Kaiser, Donato Spoltore, Hans Kleemann, Axel Fischer, Frank Ellinger, Koen Vandewal, Karl Leo

NCOMMS-20-08959-T, April 21st, 2020.

Dear reviewers,

We thank you for your careful peer-review of our manuscript and the very helpful criticism. In response to the reviewers comments, we have made substantial changes to the manuscript. In the document below we comment on all the concerns and points raised. For better readability, we visually structured our response as following:

The comments and points of the referees are copied in black color.

Our response is colored in green.

Furthermore, we cite parts from the revised manuscript or the revised supporting information to afford a quick impression of the additions and improvements to the reviewers. These citations are indicated by blue text and are shifted to the right.

We hope that we satisfactorily address all points and would appreciate if you consider manuscript ready for publication.

Sincerely,

Jonas Kublitski and coauthors

Reviewer #1 (Remarks to the Author):

This communication reports on the origin of dark reverse saturation current of OPDs based on systematic analyses. Supplementary Note 1-3 show the author's effort to objectively explain the cause of dark current. After carefully reading this communication, I feel the suggested correlation between the effective charge-transfer state energy and measured dark current is very reasonable. I strongly recommend publication of this manuscript. Nonetheless, the following points should be addressed before the final acceptance.

The authors thank the reviewer for the positive feedback and for the interesting points raised, which we address below:

1. *From the author's findings on J_D vs E_{CT} and E_{CT} vs trap states, it seems reasonable to assume that larger E_{CT} can lead to higher density of trap states. If so, what would be clear explanation on this from material side?*

From data on different material / D-A systems, a correlation between E_{CT} and the amount of trap states indeed seems to exist. Moreover, the scaling of the amount of traps with the donor concentration in TPDP devices suggests an interfacial character of the traps measured in our work, which we could speculate to be related to charge-transfer states., Determining the exact origin of the trap states will however require further research on a larger dataset to draw clear conclusions in this respect. We therefore opt not to speculate extensively on the exact origin of the trap states.

2. *Though impedance analyses fitted with Gaussian function were quite successful on analyzing trap distribution, there have been many reports that showed success of exponential distribution. The authors need to explain why they chose Gaussian. (PRB 84, 195209, 2011).*

While we agree that exponential distributions have been observed in many material systems, this was not the case for the bulk heterojunctions studied in this work. Within the energy range that we were able to probe with impedance spectroscopy, the shape of the measured trap density of states was well represented by a peaked Gaussian distribution (see for example figure 3f). Moreover, the presence of a Gaussian distribution of traps has been reported using different techniques, such as *thermally stimulated currents*¹, *CF spectral modeling*², *photo-thermal deflection spectroscopy*³, and *impedance spectroscopy*⁴. This encouraged us to describe the trap density of states with a Gaussian distribution, which is justified in the main text from lines 352 to 356.

For all of them, N_t ranges from 10^{15} cm^{-3} to 10^{16} cm^{-3} , obtained from the fit of the measured spectra to a Gaussian distribution, indicating a rather general trend for donor:C₆₀ BHJs, see Supplementary Figures 13 and 14. While many reports describe an exponential density of trap states, for the materials investigated within this work and the energy range probed by Impedance Spectroscopy, the measured spectra resemble a Gaussian distribution (fig 3f), which is in

agreement with previous investigations in organic systems based on different preparation techniques.

3. *If the origin of dark current is thermal release of trapped carriers as the authors argued, I guess shallower traps would have more dominant contribution to dark current than the case of deep traps. It would be helpful if the authors explain this point.*

Indeed, it is intuitive to think that a thermally activated process would be more efficient as trap energies get shallower and, therefore, closer to the respective transport energy level. While this is the case, the Shockley-Read-Hall generation, as well as recombination, relies on two thermally activated processes. In order to contribute to J_D an electron has to be firstly captured from the valence band and then released to the conduction band in a second thermally activated process. The second process becomes more efficient as the trap energy gets closer to the conduction band. However, concomitantly to that, the capture rate decreases exponentially, which ultimately decreases the final amount of charges released. In fact, it is possible to show that SRH finds its highest generation rate when the trap energy equals midgap^{5,6}. To clarify, we added the following explanation on line 247 to line 252.

An electron is firstly excited from the valence band to the trap state, followed by a second excitation, which releases it to the conduction band. A charge contributes to J_D only after a double thermal excitation and the final rate depends on both activation energies. As schematically shown in Supplementary Figure 7, if r_2 increases as E_t approaches the conduction band, r_4 decreases concomitantly. In fact, the highest generation rate is achieved around E_i , decreasing exponentially towards valence and conduction band, see Supplementary Note 4 for more details.

4. *The authors introduced Poole-Frenkel model to explain field dependence of the measured dark current. In this case, the assumption of even distribution of electric field along with the thickness direction of the active layer is required. However, in this report, three different layers of p, i and n with different resistances are located along with the thickness direction. The authors need to give reasonable explanation on this issue to use equation 5.*

In the simulation, the field is calculated locally at each discretization point, therefore, the assumption of evenly distributed field is not necessary. We clarified this detail in line 265.

By implementing the modified trap depth for calculating the SRH efficiency in a single-layer drift-diffusion simulation with ohmic contacts, where F is calculated locally, taking the internal charge distributions into account...

5. *Some minor points:*

- Page 2, line 52: *The authors argued that the ideal dark saturation current is thermally*

activated. However, the reverse saturation current of conventional PN junction diode is limited by both diffusion current and thermal generation. Please give reasonable modification on this. (S. M. Sze and Kwok K. NG, Physics of Semiconductor Devices).

Indeed, the sentence was not correct. The text has been fixed in lines 49 to 51.

In an ideal diode, in addition to the diffusion current, the dark saturation current (J_0) comprises a thermally activated component as a result of thermal generation of charges over the gap of the material,...

- Page 5, line 134: The authors mentioned that low donor content BHJs have been chosen to ensure minimum morphological effects. Relevant references are required here for researchers who are not familiar with molecular semiconductors.

Thanks for pointing this out. Additional references have been added in the respective part of the text, line 130.

Here, low donor content bulk heterojunctions (BHJ) have been chosen to ensure a comparable morphology, which is known to depend on D-A mixing ratio^{7,8}, miscibility⁹ and aggregation properties¹⁰...

- It would be better to explain why the authors limited the sweep range of J-V characteristics to -1.5 V -1.5 V. Note that CMOS image sensors require much higher sweep range to -5 V.

Depending on the doping concentrations of p and n layers, the device architecture and the material, such thin devices can reach the breakdown regime at around -5 V^{11,12}. In this regime other processes might dominate reverse dark current, rather than the trap-assisted generation discussed in this work. Therefore, we limited our measurement range from -1.5 V to 1.5 V. Nonetheless, in Supplementary Figure 20 we added a measurement with a larger range for three material systems and included a fit according to Murgatroyd, which considers the Poole-Frenkel effect.

Supplementary Figure 1 | *JV* characteristics of three different devices. The devices show an increase in reverse dark current which can be fitted with the analytical solution proposed by Murgatroyd *et al.*¹³ showing a dependence on the square root of the applied field. This is a further indication that Poole-Frenkel effect is present in these systems.

Reviewer #2 (Remarks to the Author):

In this paper Kublitski and co-workers report a quantitative model describing the role of mid-gap trap states as the key source of noise in organic photodiodes. This is important because, as a result of these traps, organic photodiodes show high noise spectral density (S_n), which limits their specific detectivity to around 10^{13} Jones. Although the physics of these interfacial trap states in donor-acceptor organic bulk-heterojunctions have been well-characterized in the past for solar cells, their role in limiting the photodetection performance at reverse bias operation is not fully understood. The comprehensive understanding provided by this study will promote the development of organic photodiodes with high detectivity levels, although the origin of these traps remains unknown. This manuscript is highly technical and will be of particular interest to researchers specialized on organic electronics and photodetector research. It is overall well written, but the clarity can be improved especially for non-specialists. Below are some additional comments:

We thank the reviewer for the positive feedback and for the interesting points raised. We worked to improve the clarity of the text and hope that we have achieved a more comprehensive explanation of the phenomena discussed in the paper. Below, we address the comments.

- I am confused about the description of Fig. 4a (Line 253-265). The poor fit between simulated current and experimental data in the forward bias region is attributed to the

thickness of the EBL. In that case is it possible to get a better fit using a thicker EBL in the simulation or does that lead to poor fitting at reverse bias? Simulated results for various EBL thickness should be provided and compared with experimental data in Fig 4a and Fig. S14.

For simplicity and due to time constraints, the simulations are performed for a single layer device with Ohmic contacts. Unfortunately, in Supplementary Table 7 and Supplementary Table 8 the information about EBL and HBL was somewhat unclear, which we adjusted in the new version of the manuscript. In real devices, the electric field drops along the intrinsic active layer but also along the undoped EBL and HBL. In order to properly describe the electric field, an equivalent thickness for the intrinsic layers (active layer + EBL + HBL) is used. On the other hand, it is incorrect to account the EBL and HBL for generation since the experimental results show that traps are found in the active layer. Therefore, we consider generation only in the active layer.

The correct modelling of energy barriers at interfaces in multilayer organic devices is very challenging. Within the scope of the present paper and our personnel resource, we do not see a reasonable opportunity to achieve a full description. However, we have carefully evaluated the point made by the reviewer and believe that, despite the fact that a better fit in forward direction would be desirable, the conclusion of the paper are not invalidated by this deviation: From our experimental data, we know that N_t scales with J_D demonstrating that at reverse bias, the energy barrier plays a minor role, as already pointed out by other authors.¹⁴ This leads us to conclude that our simple single-layer model is able to describe the behavior of J_D in the reverse region with sufficient accuracy. In Supplementary Table 7 and Supplementary Table 8 we describe our approach. In Line 265, we added that a single-layer model is used.

By implementing the modified trap depth for calculating the SRH efficiency in a single-layer drift-diffusion simulation with ohmic contacts, where F is calculated locally, taking the internal charge distributions into account.

- Figure 4b: does the simulated current (red and orange dashed lines) reproduce the dip in current at short-circuit?

Yes, the simulation is capable of reproducing it, as can be seen in Fig. 4a, for instance. However, at zero bias, reaching equilibrium is quite time consuming. We therefore opted for skipping this single voltage point because any experimental value at zero bias under dark represents nothing but the noise of the source meter used and has no relevant meaning here.

- Some mistakes found in the reference list: e.g. Ref. 30 and Ref. 50.

Thank you for spotting these mistakes, we updated the reference list.

Reviewer #3 (Remarks to the Author):

In a combined experimental and modeling study Kublitski et al. show that electronic traps at the donor-acceptor interfaces (so in the bulk) affect the shot noise and dark current in organic photodetectors. Though this idea is not novel, they show for the first time experimentally (using impedance measurements) that a great variety of donor-acceptor blends have low trap densities (in the range of 10^{15} - 10^{16} cm⁻³) and that these trap states are very likely associated with the (volumetric) heterojunction formed between donor and acceptor material. The microscopic origins of the traps are not discussed other than in general terms of 'defects'.

Studying dark current in these devices implies a lot of control experiments. Most of these control experiments, albeit not all, have been performed. Doing this for a dazzling amount of material systems means even more work. Especially the experiment in which the TPDP-to-C₆₀ ratio was systematically varied, is elegant, and the observed relationship between N_t and dark current insightful. My compliments!

The vast magnitude of material systems studied and large number of experimental techniques used, however, also makes the manuscript difficult to read (and write). Not all data can be shown but I sometimes fail to understand the selection made by the authors.

The discussion and interpretation of the impedance results should be improved. Using the fitting parameters of the trap states J_D is calculated using a relatively simple drift-diffusion model. Given my comments below on the experimental accuracy of the fitting, more simulation data can help to get better physical insights. Showing that the simulations and experimental results are consistent is a rather meager conclusion, I find. Authors should perform a more thorough analysis on what the results tell us, what can be properly explained and what not.

We thank the reviewer for the honoring our work and the outstanding quality of the criticism, which was very helpful to improve our manuscript. In the first version of this manuscript, we did indeed not go in large depth into the characterization methodology, which might have affected the clarity. In this updated version of the manuscript, we tried to clarify our analysis, based on the suggestions made by the reviewer.

Fundamental points

Measuring low density of trap states is very difficult, so the interested reader should be able to follow the argumentation and analysis of the authors step-by step.

1. The analysis of thermal admittance spectroscopy to derive the properties of trap states in low-mobility semiconductor devices is debated in literature. See for instance: H.S. Pang et al. Capacitive methodology for investigating defect states in energy gap of organic semiconductor, Organic Electronics 2018; F. Werner et al. Can we see defects in capacitance measurements of thin-film solar cells; S. Wang et al. Understanding thermal admittance spectroscopy in low-mobility semiconductors, J Phys Chem 2018. This paper should include a proper discussion on the use of impedance spectroscopy, incl. its limitations.

We agree with the reviewer that it is difficult to measure low density of trap states; a discussion on the mentioned points was added from line 201 to 211. A discussion on the limitations of the characterization method is included in the Supplementary Note 3.

Measuring traps in organic solids is a rather difficult task due to the lack of simple techniques able to access their concentration and energetic characteristics. Nonetheless, a widely applicable method was proposed by Walter *et al.* based on the capacitive response of these states, whose occupation varies with the modulation of the signal. On the one hand, this enables the characterization of defects by a straightforward technique, such as impedance spectroscopy (IS). On the other hand, such a technique probes many different effects in the device, especially in an organic one, which can show identical capacitive spectra to measurement can in fact reveal the trap characteristics¹⁵⁻¹⁷. Special caution should be taken when using this method for low mobility materials and in devices where energetic transport barriers are present. As both are the case in our devices, in Supplementary Note 3 we discuss the limitations of the method and show that for our devices, IS shows meaningful results.

Supplementary Note 3. Impedance Spectroscopy in Organic Blends

Using the method introduced by Walter *et al.* to characterize trap states organic devices is debated in literature, especially when dealing with low mobility materials or devices where energy barriers are present. As both conditions apply for our devices, we exemplarily compare different devices and analyze the effects on the trap characterization to ensure that the results discuss in the main text can be accurately estimated. In order to do that, we analyze devices comprising different thicknesses and under different biasing conditions.

Supplementary Figure 5, contains data for devices with different active layer thicknesses. The trap density is not expected to depend/vary with thickness. Indeed, from 50 to 150 nm, N_t remains constant. As indicated in the main text, the attempt-to-escape frequency, ν_0 , is obtained by overlapping N_t measured at different temperatures. For 150 nm, in order to achieve that, ν_0 has to be set to a lower value. As pointed out by different research groups, thicker devices¹⁵ and low mobility materials¹⁷ lead to a wrong estimation of ν_0 . This further explains why using Equation S2.1 as a direct estimation of the recombination rate is not possible as ν_0 can be underestimated. Therefore, the values of ν_0 must be taken only a first approximation in our study, representing a limitation of the method. Also, E_t can be slightly affected, depending on the mobility of the blend. The results shown in Supplementary Figure 5 were measured in devices fabricated in the same batch, but in a different batch than that of the samples presented in the main text, explaining the small deviations in the absolute amount of traps.

Supplementary Figure 2 | Trap density for TPDP:C₆₀ (13.3 mol%) with different device thicknesses.

Another important aspect when applying this method in devices is the presence of energy barriers, as they can produce the same signature in the capacitance spectra as those produced by traps. As argued by Siebentritt *et al.*, the occupancy of trap states is governed by the crossing of the Fermi level with the trap level, therefore, any trap signature should disappear at high enough forward bias, since the Fermi level will no longer cross the trap level¹⁶. Following the same reasoning, a minority carrier trap signature should also disappear at high enough reverse bias. Indeed, measuring our device at different biases we can clearly observe this effect: the step in the capacitance spectra, observed at zero bias in the range from 10 Hz to 10 kHz, disappears when both forward and reverse bias are used. From this measurement, shown in Supplementary Figure 6, we can infer that the step in the capacitance arises from traps and, more importantly, that these states are minority carrier traps¹⁶.

Supplementary Figure 3 | Capacitance spectra at different biases for a device based on TPDP:C₆₀ (13.3 mol%). Note that the reconstruction of the trap density uses the derivative of the capacitance spectra. This implies that all spectra at

bias below -0.3 V, as well as above 0.3 V, lead to the same trap density, which tends to zero, given their rather constant shape.

2. *The authors should provide the ‘raw’ impedance spectroscopy data (of TPCP in Figure 3, and of the other donors in the Supplementary Information) and discuss measurement reproducibility.*

The raw data is now shown in Figure 3 at one temperature for all concentrations of TPDP and for one concentration at different temperatures. The temperature dependent data of the remaining concentration and donors are included in Supplementary Figure 12 and 14.

3. *The authors should clearly show that it is possible to distinguish between deep traps and capacitive contributions (from for instance transport layers or additional layers in the device). I miss here a simple control experiment: does N_t (and J_D for a given electric field) stay constant with increasing thicknesses of the BHJ? Can the authors comment on this? Is it experimentally difficult?*

In Supplementary Note 3, we also add a discussion of the effects of blocking layers. In addition to the thickness variation, showing a constant trap density, the method proposed by Siebentritt *et al.* allowed us to show that the step in capacitance is related to minority carrier traps¹⁶. The authors thank the reviewer for indicating this reference.

4. *The authors should elaborate on their choice of attempt-to-escape frequency as it directly relates to trap energies. Why do the authors use different values for this parameter for different donors?*

Due to the rather low concentration of traps measured in these devices, the data is noisy, making the extraction of the attempt-to-escape frequency, ν_0 , by the Arrhenius-behavior of ω_0 , difficult. Alternatively, Walter *et al.* suggested that for a distribution of traps one can choose ν_0 such that the overlap of the traps distribution at different temperatures is achieved¹⁸. This approach was used for all impedance data presented in this work. A sentence clarifying the used method was added in line 217.

...,and overlapping the data by the appropriated choice of ν_0 , we can reconstruct the trap density,...

Moreover, as now discussed in Supplementary Note 3, the estimation of ν_0 for low mobility materials is indeed a limitation of the method.

As pointed out by different research groups, thicker devices¹⁵ and low mobility materials¹⁷ lead to a wrong estimation of ν_0 . This further explains why using Equation S2.2 as a direct estimation of the recombination rate is not possible as ν_0 can be underestimated. Therefore, the values of ν_0 must be taken only a first approximation in our study, representing a limitation of the method.

5. The extracted trap densities are fitted with a single Gaussian but the quality of the fit is rather poor in most cases (Figure 10 in SI). The authors fitted trap distribution in the donor:C₆₀ with a single Gaussian distribution that has its maximum at 0.51-0.57 (Figure 3b and SI Figure 10) and an energetic disorder of 0.04-0.10 eV. Looking at the data of ZnPc and Spiro-MeO-TPD, this choice is debatable, and thus not evidently points to the conclusion stated by the authors that ‘the distribution of traps, ..., indicates a rather general trend for donor:C₆₀ BHJs (p.1, line 306)’. Authors should elaborate on the fitting procedure and the extracted parameters so the reader can assess if the conclusion is inevitable.

Indeed, due to the rather low trap densities, achieving a good description by a single Gaussian is difficult. Besides including the raw data, we defined in the main text the region where the trap analysis was made, namely from 10 Hz to 100 kHz (see lines 213 to 225). In this frequency range, the Gaussian behavior is most evident. As argued in the main text, frequencies above and below these limits might suffer from different effects, such as overcoming energy barriers and charge transport effects, which compromise the correct interpretation of the data. Moreover, in Supplementary Figure 13, we now explicitly mentioned that the fitting range is confined and that this implies that traps below or above that energy are not taken into account.

Excluding the range below 10 Hz, where the capacitance spectra are dominated by resistance of the layers, which, as expected, increases for lower concentrations, and above 10 kHz, where the geometric capacitance and the series resistance of the contacts controls the spectra, **Error! Reference source not found.**b shows an increase in the capacitance, which we attribute to traps. By studying this region in more detail at different temperatures, as depicted in **Error! Reference source not found.**c for 26.7 mol%, and overlapping the data by the appropriated choice of v_0 , we can reconstruct the trap density, shown in **Error! Reference source not found.**f for this concentration and in **Error! Reference source not found.**e one temperature per concentration, is plotted. The reconstructed trap densities for every concentration have been fitted with a single Gaussian distribution function each, similarly to **Error! Reference source not found.**f, from which N_t , E_t and the broadness of the Gaussian, σ , were extracted, see inset, Supplementary Note 2 and Supplementary Figures 11 and 12. N_t is summarized in **Error! Reference source not found.**d and compared to the absolute value of $J_D(-1.0V)$. The increase of J_D with increasing N_t suggests that extra dark current is produced by the generation of charges via trap states.

The extracted trap parameters serve as input for the simulations.

1. Even in the case of the best fit (TPDP, Figure 3b), fitting the experimental data with a single Gaussian with maximum at around 0.5 eV (roughly midgap) with a sigma of 0.04 eV, implies that traps with energies lower than 0.4 eV and higher than 0.55 eV can be neglected. The authors should show quantitatively (‘illustrate’ with simulations) how important the exact distribution of traps is in the J_σ -V simulations and thus the applicability of the model (SRH plus Poole Frenkel). It would be very convincing when

the authors can show that the fit parameters of the trap distribution can simulate the absolute magnitude of J_D , its field dependence and temperature dependence simultaneously and consistently, at least for one donor:C₆₀ system. If this is not possible, then a discussion on parameter sensitivity is required. In the current manuscript three fitting parameters are used to describe the trap distribution. Would a constant DOS not explain the J_D data (also considering that midgap states are most prolific, as seen from eq. 4)? Or what about a fit with a single Gaussian which has its energy maximum fixed at half the value of the CT energy, and N_t and sigma are the only two fit parameters? The authors are invited to simulate the alternative approaches, discuss the results and relate back to previous work of for instance Fallahpour (ref. 39 in the manuscript).

In Supplementary Figure 23, we now show simulations for different values of the broadness of the trap density of states (σ), as well as different positions of the trap distribution maximum. Within the simulated range, sigma indeed plays a minor role. In fact, it is not possible to correlate J_D with sigma. It is thus not excluded that a constant DOS, i.e. a very large sigma, could be used to obtain a similar value of J_D , similar to what was already shown by Fallahpour *et al.* for a Gaussian trap distribution centered at midgap. However, our DOS distributions have been independently obtained from the impedance spectroscopy measurements, and we use those as input to simulate both magnitude and field-dependence of J_D , as shown for three donor-acceptor / TPDP:C₆₀ concentrations in Figure 4a.

On the other hand, as expected from Eq. 4, the energetic position of the trap distribution strongly affects the final value of J_D , as shown in Supplementary Figure 23. As traps are positioned further from midgap, their contribution to J_D decreases, leading to a lower J_D . Thus, if we would assume the trap distribution at midgap (in contrast to our experiments, showing that it is centered 60 meV away from midgap), the simulation would overestimate the measured J_D . This would require adjusting other parameters in order to match J_D , which seems inconsistent with our experimental findings. It is also worth mentioning that N_t and σ are not used as fitting parameter to achieve an agreement with J_D , but rather used in the simulation as measured. The work by Fallahpour *et al.* is now considered in more details from line 271 to line 275.

As discussed by Fallahpour *et al.*, a generic trap distribution at midgap is also able to reproduce the magnitude of J_D observed in many OPDs. However, besides failing in reproducing the field dependence, such approach is debatable, given the major influence of N_t and E_t on J_D . The impacts of σ and E_t are discussed in Supplementary Figure 23.

As described from line 338 to line 342, our simulation is not able to reproduce the experimental temperature dependence. We attribute that to the many temperature-dependent effects, such as transport and overcoming of energy barriers, which are not explicitly taken into account in our drift-diffusion model.

A quantitative agreement of the absolute E_a values is however not possible, since the experimental values are affected by other thermally activated processes that

we did not take into account in the simulations, such as charge transport and overcoming energy barriers.

2. *The authors should explain if (and why) they think V_{bi} should be taken into account when simulating the current-voltage data.*

V_{bi} describes how steeply the electrical field drops across the undoped layers, therefore, affecting activation to and from the traps in the framework of the Poole-Frenkel effect. Therefore, we conclude that V_{bi} should be taken into account. For simplicity, in the simulation we assume ohmic contacts and define V_{bi} as the difference in work function of cathode and anode. This information is now mentioned in Supplementary Table 7 and Supplementary Table 8.

3. *In this work, activation energies of J_D are determined using Arrhenius-behavior, using a relatively small temperature range. The validity of this analysis is questionable. In many organic semiconductors a temperature dependence of charge transport $\sim \exp(-a/T^2)$ rather than the expected thermally activated transport $\exp(-b/T)$ is observed. Authors should discuss the meaning and implications of this.*

We included a larger temperature range, from 223.15 K to 303.15 K. In this region, we observed a linear behavior in the Arrhenius plot, as shown in Supplementary Figure 20, exemplarily extracted at -1.5 V. As the data is better describe by the relation $\exp(-b/T)$, we felt that it was not necessary to discuss this topic, as the behavior mentioned by the reviewer is not observed.

4. *Extraction barrier can influence J_D (SI Figure 14). Is this also relevant for the data shown in Figure 2, where most of the diodes seem to suffer from extraction barriers, judging from the forward currents. Please discuss possible implications (quantitatively).*

For the systems shown in Figure 2, an extraction barrier for holes under reverse bias is present for the donors TTDTP, TPDP, and m-MTDATA, for which the HOMO levels are higher (~ -4.85 eV to ~ -5.00 eV) than the HOMO level of the HTL/EBL (~ -5.10 eV). In these devices, some minor effects of an extraction barrier could be expected in the reverse region: the current could be decreased, as holes are not extracted as efficiently. For the remaining systems, no extraction barriers are present. Instead, an injection barrier for holes under forward bias is present, as the HOMO levels of the donors are lower than the HOMO level of the HTL/EBL, which explains the shape of the forward region for these systems. For the systems which are not affected in the reverse region, we see a scaling of the reverse J_D as a function of E_{CT} . Yet, even for those systems where J_D could be lowered by an extraction barrier, the same trend is observed. For these devices, in absence of such a barrier, the reverse current could only be higher, as an extraction barrier is not expected to increase the current, making the observed trend of reverse J_D as a function of E_{CT} only stronger. This suggests that extraction barriers play a minor role in this case. We discuss this issue in lines 144 to 154.

For the sake of comparison, we kept the same structure for all devices shown in **Error! Reference source not found.**, see also the inset. This implies that for high- E_{CT} combinations, from Spiro-MeO-TPD towards lower HOMOs, an injection

barrier for holes under forward bias is formed between donor and electron blocking layer (EBL: MeO-TPD). Such a barrier only affects the forward regime of the JV curves up to the space-charge limited current region. For low- E_{CT} combinations, having TTDTP, TPDP or m-MTDATA as donor, an extraction barrier under reverse bias is formed at the same interface. This barrier can only decrease the reverse J_D , as holes are prevented from being extracted. However, even for these systems, the same trend of J_D versus E_{CT} is observed. In the absence of extraction barriers, this relation between E_{CT} and J_D could only be strengthened. The extraction barrier also affects the forward region of the the JV curves, which is reflected in **Error! Reference source not found.** and will be discussed later in more detail.

Suggestions for further improvements:

5. Figure 1a illustrates that experimental values of specific detectivity are much larger than the intrinsic values, as found in this paper as well as previous papers. The results of two of the 6 donor materials studied in this work are mentioned explicitly. Figure 1b shows the noise measurements of 4 of the 6 donor materials. To me these choices look random, also when considering that in the remainder of the text most work is done on the TPDP:C₆₀ system. I suggest the authors to elude on their choices, include data of all 6 donor:C₆₀, or be more consistent in another way (and perhaps take out the literature data in Figure 1a, it does not add much more than already mentioned in the text.) I could not find the original references to the D* values of the inorganic PDs in Figure 1a.

Figure 1a was designed to present the main problem that OPDs currently face to the reader. ZnPc and P4-Ph4-DIP were picked for being a medium and high E_{CT} system and, among the studied systems, to better represent the current range of EQE in organic materials. The TPDP:C₆₀ system has a low EQE (maybe some numbers). This also means that for this system, the contribution of achieving an EQE of 100% to the detectivity would be more important as compared to a hypothetical device with an ideal dark current at the experimentally obtained EQE. We believe that TPDP:C₆₀ would not exemplify our motivation in Figure 1a, since it does not represent the state-of-the-art of EQE in organic materials.

In **Error! Reference source not found.**a, this issue is visualized for two of the systems studied in this work: the role of EQE and S_n are compared for P4-Ph4-DIP:C₆₀ and ZnPc:C₆₀, which, besides having representative CT energies, better represent the state-of-art EQE of current diode based OPDs.

In Figure 1b, we show four of the studied systems, including the two systems discussed in Figure 1a and TPDP, as this is the main characteristic which will be discussed within the rest of the paper. For clarity, the remaining materials are presented in the SI.

Information about inorganic PDs was added, thank you for noticing our mistake.

6. Rather than the 2008 paper of Ramuz, include the more recent paper of Biele et al. (*Adv. Mater. Technol.* 2019, 4, 1800158) or alternative reports that reflects today's State of Art much better.

Indeed the work by Biele et al.¹⁹ is more representative for the current development of OPDs, thank you for sharing this reference. The reference was included.

7. Introduce the meaning of I_{bias} when discussing the noise measurements.

We introduced the definition of I_{bias} in line 97.

...level, calculated as $\sqrt{2qI_{\text{bias}}}$, where I_{bias} represents the current driven through the device at a bias voltage at -0.8 V.

8. Please, look carefully at the colors of the different J-V curves in Figure 2a. I find it difficult to see what forward currents correspond to what reverse currents. Please discuss why forward current density is low in some of these devices.

We included symbols also in forward direction to guide the eye of the reader. The effect of energy barriers is discussed from line 144 to line 154, to which we attribute the low current density.

For the sake of comparison, we kept the same structure for all devices shown in **Error! Reference source not found.**, see also the inset. This implies that for high- E_{CT} combinations, from Spiro-MeO-TPD towards lower HOMOs, an injection barrier for holes under forward bias is formed between donor and electron blocking layer (EBL: MeO-TPD). Such a barrier only affects the forward regime of the JV curves up to the space-charge limited current region. For low- E_{CT} combinations, having TTDTP, TPDP or m-MTDATA as donor, an extraction barrier under reverse bias is formed at the same interface. This barrier can only decrease the reverse J_{D} , as holes are prevented from being extracted. However, even for these systems, the same trend of J_{D} versus E_{CT} is observed. In the absence of extraction barriers, this relation between E_{CT} and J_{D} could only be strengthened. The extraction barrier also affects the forward region of the the JV curves, which is reflected in **Error! Reference source not found.** and will be discussed later in more detail.

9. Authors mention two times (on p.5 and p.6) that the observed scaling of J_{D} with E_{CT} is in agreement with equation 2. It is not! Equation 2 predicts an exponential dependence. This is different from the $J_{\text{D}}-E_{\text{CT}}$ behavior plotted in Figure 2b.

Indeed, Eq. 2 predicts an exponential dependence, while the experimental data does not show such a trend; besides it scales with E_{CT} . We modified the text to describe the comparison more accurately, see lines 137 and 162.

... E_{CT} , however, not as predicted by Equation **Error! Reference source not found.**, from which an exponential dependence is expected...

Despite these low dark currents and the observed scaling of $J_{\text{D}}(-1 \text{ V})$ with E_{CT} ,...

10. Ref 23 and 24 deal with the relationship between V_{oc} and energy difference between HOMO of the donor and LUMO of the acceptor, but do not discuss it in the framework of dark current and photodetectors. A recent paper of Simone et al. (On the Origin of Dark

Current in Organic Photodiodes, *Adv. Optical Mat* 2019) does, so might be worth to include that one instead of 23 and 24.

The reference was included. We kept reference 23, as this was the first work to mention a possible relation between J_0 and the HOMO-LUMO difference²⁰.

11. Would be good to include a reference to a recent paper of Zhang et al. *Sequentially deposited versus conventional non-fullerene organic solar cells*, *Adv. Energy Materials*, as this work also uses impedance spectroscopy to analyze interface trap states in BHJ, and find trap states centered between 0.5-0.6 eV of 10^{17} cm^{-3} .

The reference was included.

12. The 'dark' ideality factor of the J-V curves in Figure 3a decreases with increasing donor content in the film. This ideality factor is often associated with traps and recombination processes but this is not done in the present manuscript. Can the measured trap states explain the evolution of the dark ideality factor quantitatively?

We included a discussion about the ideality factor in these devices and the relation with the energy barrier, see lines 305 to 316. In addition, Supplementary Figure 21 was added where we analyze the ideality factor of the TPDP devices.

It is common to associate the ideality factor, i.e., the slope of the exponential region of the JV curve (n_{id}) to recombination processes, and it should approach two when trap-assisted recombination dominates. From a first analysis, one could conclude that as the trap concentration increases, in **Error! Reference source not found.**, n_{id} decreases. However, because of the extraction barrier, the information given by n_{id} is misleading and its analysis nontrivial. In fact, by locally accessing n_{id} at different temperatures, we can define a more meaningful region where n_{id} can be characterized²¹. As shown in Supplementary Figure 21, n_{id} is between 1.7 and 1.8 for all devices, indicating that trap-assisted recombination is the dominating process. However, no clear trend with the amount of traps could be observed, which can still be a result of the extraction barrier, whose effect seems to get more pronounced for high concentration devices. This observation is in agreement with Supplementary Figure 10, where E_{CT} slightly decreases with the TPDP concentration, indicating that the extraction barrier increases.

13. Simple Poole-Frenkel predicts a \sqrt{E} dependence. This is frequently observed in organic devices. Please indicate if J_D scales with applied electric field in these devices? If not, elaborate on this.

In Supplementary Figure 20, we show the fit according to the model proposed by Murgatroyd¹³ for three devices. The reverse current fits very well to the model for a large voltage range.

14. Authors promise us new optimization pathways to reduce noise current (p.3 line 71) but don't come back with clear recommendations. Please add, or take out this part in the introduction.

We rephrased the mentioned sentence (now lines 67 to 70) to better define what we meant and added a brief comment on p.13, line 360. We believe that the relation between J_D and traps allows the reader to refocus the optimization routine on material properties such as purity, structural defects or even on the source of such trap states, rather than device engineering, which, as shown for these materials, did not improve J_D significantly. On the other hand, E_{CT} provides a fundamental limit to seek for. If that is achieved, the current discussion on reducing voltage losses on organic solar cells will also apply for organic photodetectors, as they are related via Eq. 2 and Eq. 3.

The discovery of the relations between mid-gap traps and J_D , reported in this paper, refocuses the current optimization routines, targeting material properties rather than device engineering. Moreover, E_{CT} determines the thermal lower limit of J_D to seek for and provides a metric for judging how far J_D is from this fundamental limit.

15. *Supplementary p3. L 70 'It states that 'this indicates that, in the case of TPDP, the intrinsic processes happening in the blend dominate J_D and the minor effect of selectivity is no longer observed. Following formal logic this first conclusion cannot be drawn from the sole observation that J_D does not reduce by using p-HATNA-Cl₆ in case of TPDP'. Please rephrase.*

The respective sentence was rephrased.

Interestingly, the same is not true for the TPDP BHJ. This indicates that, in the case of TPDP, another mechanism dominates J_D and the minor effect of selectivity is no longer observed.

References:

1. Nikitenko, V. R., Heil, H. & Von Seggern, H. Space-charge limited current in regioregular poly-3-hexyl-thiophene. *J. Appl. Phys.* **94**, 2480–2485 (2003).
2. Burtone, L., Fischer, J., Leo, K. & Riede, M. Trap states in ZnPc:C60 small-molecule organic solar cells. *Phys. Rev. B - Condens. Matter Mater. Phys.* **87**, 1–8 (2013).
3. Kuik, M. *et al.* Optical detection of deep electron traps in poly(p-phenylene vinylene) light-emitting diodes. *Appl. Phys. Lett.* **99**, 2009–2012 (2011).
4. Zhang, J. *et al.* Sequentially Deposited versus Conventional Nonfullerene Organic Solar Cells: Interfacial Trap States, Vertical Stratification, and Exciton Dissociation. *Adv. Energy Mater.* **9**, 1902145 (2019).
5. Pieters, B. E., Decock, K., Burgelman, M., Stangl, R. & Kirchartz, T. Advanced Characterization Techniques for Thin Film Solar Cells. in *Advanced Characterization Techniques for Thin Film Solar Cells* (eds. Abou-Ras, D., Kirchartz, T. & Rau, U.) 633–659 (Wiley-VCH Verlag GmbH & Co. KGaA, 2016).
6. Sze, S. M. & Ng, K. K. *Physics of Semiconductor Devices*. (2007).

7. Vandewal, K. *et al.* Absorption tails of donor:C60blends provide insight into thermally activated charge-transfer processes and polaron relaxation. *J. Am. Chem. Soc.* **139**, 1699–1704 (2017).
8. Vandewal, K., Himmelberger, S. & Salleo, A. Structural factors that affect the performance of organic bulk heterojunction solar cells. *Macromolecules* **46**, 6379–6387 (2013).
9. Ye, L. *et al.* Quantitative relations between interaction parameter, miscibility and function in organic solar cells. *Nat. Mater.* **17**, 253–260 (2018).
10. Che, X. *et al.* Donor–Acceptor–Acceptor’s Molecules for Vacuum-Deposited Organic Photovoltaics with Efficiency Exceeding 9%. *Adv. Energy Mater.* **8**, 1–6 (2018).
11. Kleemann, H. *et al.* Organic zener diodes: Tunneling across the gap in organic semiconductor materials. *Nano Lett.* **10**, 4929–4934 (2010).
12. Kleemann, H. *et al.* Reverse breakdown behavior in organic pin-diodes comprising C60 and pentacene: Experiment and theory. *Org. Electron.* **14**, 193–199 (2013).
13. Murgatroyd, P. N. Theory of space-charge-limited current enhanced by Frenkel effect. *J. Phys. D. Appl. Phys.* **3**, 151 (1970).
14. Fallahpour, A. H., Kienitz, S. & Lugli, P. Origin of Dark Current and Detailed Description of Organic Photodiode Operation Under Different Illumination Intensities. *IEEE Trans. Electron Devices* **64**, 2649–2654 (2017).
15. Xu, L., Wang, J. & Hsu, J. W. P. Transport Effects on Capacitance-Frequency Analysis for Defect Characterization in Organic Photovoltaic Devices. *Phys. Rev. Appl.* **6**, 1–10 (2016).
16. Werner, F., Babbe, F., Elanzeery, H. & Siebentritt, S. Can we see defects in capacitance measurements of thin-film solar cells? *Prog. Photovoltaics Res. Appl.* **27**, 1045–1058 (2019).
17. Wang, S., Kaienburg, P., Klingebiel, B., Schillings, D. & Kirchartz, T. Understanding Thermal Admittance Spectroscopy in Low-Mobility Semiconductors. *J. Phys. Chem. C* **122**, 9795–9803 (2018).
18. Walter, T., Herberholz, R., Müller, C. & Schock, H. W. Determination of defect distributions from admittance measurements and application to Cu(In,Ga)Se₂based heterojunctions. *J. Appl. Phys.* **80**, 4411–4420 (1996).
19. Biele, M. *et al.* Spray-Coated Organic Photodetectors and Image Sensors with Silicon-Like Performance. *Adv. Mater. Technol.* **4**, 1–6 (2019).
20. Potscavage, W. J., Yoo, S. & Kippelen, B. Origin of the open-circuit voltage in multilayer heterojunction organic solar cells. *Appl. Phys. Lett.* **93**, 193308 (2008).

21. Wu, J., Fischer, A. & Reineke, S. Investigating Free Charge-Carrier Recombination in Organic LEDs Using Open-Circuit Conditions. *Adv. Opt. Mater.* **7**, 1–10 (2019).

Reviewer #1 (Remarks to the Author):

The revised manuscript and response letter answered well to the concerns raised. And the requested modifications were well reflected in the manuscript. In my opinion, the manuscript is now ready to be accepted.

Reviewer #2 (Remarks to the Author):

The authors have addressed the points raised by the reviewers and have revised the manuscript accordingly. I believe this manuscript can be accepted for publication.

Reviewer #3 (Remarks to the Author):

The authors fully addressed my comments and improved the manuscript further. I am especially satisfied with how the authors extensively address the impedance results and analysis.